# Proceedings of the 5th Asia Dengue Summit

**DOI:** 10.3390/tropicalmed8040231

**Published:** 2023-04-19

**Authors:** Nattachai Srisawat, Duane J. Gubler, Tikki Pangestu, Usa Thisyakorn, Zulkifli Ismail, Daniel Goh, Maria Rosario Capeding, Lulu Bravo, Sutee Yoksan, Terapong Tantawichien, Sri Rezeki Hadinegoro, Kamran Rafiq, Valentina Sanchez Picot, Eng Eong Ooi

**Affiliations:** 1Tropical Medicine Cluster, Chulalongkorn University, Bangkok 10330, Thailand; 2Center of Excellence in Critical Care Nephrology, Faculty of Medicine, Chulalongkorn University, Bangkok 10330, Thailand; 3Excellence Center for Critical Care Nephrology, King Chulalongkorn Memorial Hospital, Bangkok 10330, Thailand; 4Academy of Science, Royal Society of Thailand, Bangkok 10330, Thailand; 5Program in Emerging Infectious Diseases, Duke-NUS Medical School, Singapore 169547, Singapore; 6Yong Loo Lin School of Medicine, National University of Singapore, Singapore 169547, Singapore; 7Faculty of Tropical Medicine, Mahidol University, Bangkok 10330, Thailand; 8Department of Pediatrics, KPJ Selangor Specialist Hospital, Malaysia; 9Division of Pediatric Pulmonary Medicine and Sleep, Khoo Teck Puat National University Children’s Medical Institute, National University Hospital, Singapore 169547, Singapore; 10Research Institute for Tropical Medicine, Muntinlupa City 1781, Philippines; 11College of Medicine, University of the Philippines Manila, Manila 1000, Philippines; 12Center for Vaccine Development, Institute of Molecular Biosciences, Mahidol University, Nakhon Pathom 73170, Thailand; 13Division of Infectious Diseases, Department of Medicine, Chulalongkorn University, Bangkok 10330, Thailand; 14Department of Child Health, Faculty of Medicine, Universitas Indonesia, Jakarta 10430, Indonesia; 15International Society of Neglected Tropical Diseases, London WC2H 9JQ, UK; 16Fondation Merieux, 69002 Lyon, France

**Keywords:** dengue, Asia, strategies, dengue control, vaccine update, therapeutic targets, innovations

## Abstract

The 5th Asia Dengue Summit, themed “Roll Back Dengue”, was held in Singapore from 13 to 15 June 2022. The summit was co-convened by Asia Dengue Voice and Action (ADVA), Global Dengue and Aedes transmitted Diseases Consortium (GDAC), Southeast Asian Ministers of Education Tropical Medicine and Public Health Network (SEAMEO TROPMED), and the Fondation Mérieux (FMx). Dengue experts from academia and research and representatives from the Ministries of Health, Regional and Global World Health Organization (WHO), and International Vaccine Institute (IVI) participated in the three-day summit. With more than 270 speakers and delegates from over 14 countries, 12 symposiums, and 3 full days, the 5th ADS highlighted the growing threat of dengue, shared innovations and strategies for successful dengue control, and emphasized the need for multi-sectoral collaboration to control dengue.

## 1. Introduction

The 5th Asia Dengue Summit (ADS), co-convened by Asia Dengue Voice and Action (ADVA), Global Dengue and Aedes transmitted Diseases Consortium (GDAC), Southeast Asian Ministers of Education Tropical Medicine and Public Health Network (SEAMEO TROPMED), and the Foundation Mérieux (FMx), was held in Singapore from 13 to 15 June 2022. The 5th ADS, themed “Roll Back Dengue”, included dengue experts from academia and research and representatives from the Ministries of Health in several Asian countries, the Regional and Global World Health Organization (WHO), and the International Vaccine Institute (IVI).

The summit emphasized that dengue control requires strong multi-sectoral collaboration between health, environment, and education ministries with cooperation from the private sector and communities. Sustained vector control, ongoing surveillance, vaccine advocacy, political commitment, public awareness and social mobilization, healthcare capacity building, and continuous research and innovation are key components of dengue preparedness across the Asia-Pacific region. 

With more than 270 speakers and delegates from over 14 countries, 12 symposiums, and 3 full days of enriching in-depth dengue discussions, the 5th Asia Dengue Summit provided an ideal platform for dengue stakeholders to share best practices and strategies to fight against dengue. The three-day summit included presentations and discussions on evolution in dengue epidemiology, the role of antivirals in dengue prevention, advances in vector control and surveillance, vector biology and ecology, innovations in diagnostics, vaccines, and therapeutics, and the role of data sciences in dengue prevention and control. The key highlights from the 5th Asia Dengue Summit are presented here.

## 2. Monitoring Dengue Epidemiological Trends 

Asia bears 70% of the global dengue burden [1], with Southeast Asia recording the highest age-standardized incidence rates with an annual average of 34.3 cases per 1000 people and the highest dengue mortality rates (8.49 cases per million in 2013) [2]. Regular extinction of existing strains, the emergence of new strains, the introduction of new strains from other regions, and the local evolution of the dengue virus (DENV) are instrumental in the development of dengue epidemics [3]. The intriguing landscape of changing DENV serotypes and genotypes in Southeast Asia (SEA) emphasizes the evolving dengue epidemiology. A massive outbreak of DENV1 that occurred in Northern Vietnam in the year 2017, with 36,345 DENV1 cases in Hanoi, provides an illustration of the role of locally circulating DENV strains in causing dengue outbreaks. Phylogenetic analysis of blood samples collected during the large-scale 2017 DENV1 outbreak in Northern Vietnam showed that the coding sequences were part of a single cluster of sequences from 2009 and 2016, suggesting that the 2017 outbreak was caused by a locally circulating virus, highlighting the challenges in the control of locally circulating viruses [4]. The Philippines provides another interesting example of the evolution of the dengue virus to increase transmission. The two DENV-4 strains (GI and GIIa) isolated during the 2015–2017 epidemic were found to be closely related to distinct groups of GIIa strains present in the Philippines since 2004, illustrating the genetic evolution of the dengue virus to sustain transmission. This rapid evolution of DENV-4 with the disappearance of the GI strain and a genotype shift toward the GIIa strain appears to be exclusive to the Philippines since it is different from the global GIIa strains [5]. In another such instance from Myanmar, where DENV1–4 strains have been circulating for several years, the emergence of the new DENV-3 genotype-I (GI) coincided with the increase in the number of DENV-3 cases between 2017 and 2019, emphasizing the evolution of dengue virus to increase transmission efficiency [3]. The continuous evolution of the DENV serotype and genotype and their role in developing dengue outbreaks emphasize the need for consistent epidemiological surveillance for effective dengue control. These examples highlight the importance of tracking dengue epidemiological trends and dengue virus evolution to plan dengue control measures and monitor the progress of preventive efforts. Though the global COVID-19 pandemic temporarily influenced dengue epidemiology in many countries, without further extensive lockdowns or travel restrictions, dengue incidence is expected to revert to pre-pandemic levels. Consequently, there is a collective need for more effective and sustainable tools for monitoring dengue epidemiological trends.

## 3. Improving Clinical Vigilance

Though dengue was originally considered a disease affecting children, there has been a gradual shift in the clinical profile from the pediatric to the adult population. However, it is harder to recognize and diagnose dengue early in older adults as the symptoms may not be typical. Nevertheless, older adults are at higher risk of severe outcomes due to a higher frequency of pre-existing co-morbidities and compromised organ function and thus less reserved to withstand the insult from dengue. Atypical presentation and absence of classic dengue symptoms can delay the clinical diagnosis in older adults [6]. Since myalgia, arthralgia, retro-orbital pain, and mucosal bleeding are less common in older patients, the presence of fever and leukopenia in the absence of other symptoms should prompt suspicion of dengue [7]. There is a higher incidence of gastrointestinal bleeding, acute renal failure, pleural effusion, and longer hospitalization due to dengue in older adults. Underlying conditions such as rheumatoid arthritis, stroke, chronic kidney disease, liver disease, asthma, coronary artery disease, and chronic obstructive pulmonary disease increase the risk of hospitalization and intensive care unit admission [8]. The association between underlying chronic non-communicable diseases (NCDs) and adverse outcomes in dengue cannot be overemphasized. The growing prevalence of NCDs and the epidemiological shift in dengue affecting older adults indicates that the dengue disease burden includes complex cases with underlying co-morbidities. Consequently, the management of older patients is challenging, requiring judicious fluid administration, maintenance of cardiac function, and careful consideration of underlying chronic conditions and medications [9].

## 4. Understanding Chronic Post-Dengue Sequelae

While dengue is mostly considered an acute illness, adverse effects on quality of life are seen in some patients, even after recovery. Some patients have reported persistent symptoms, particularly depression, fatigue, and weight loss after recovery. Such persistent symptoms result in a loss of productivity and an increase in medical expenses. Though the exact mechanism of chronic sequelae is debated, the complex interaction between neuroendocrine, musculoskeletal, and immunological systems may be responsible for persistent fatigue [10]. The true incidence, duration, and severity of post-dengue chronic sequelae remain poorly defined. A recent prospective study showed that almost 20% of dengue patients continued to experience central nervous system-related symptoms, such as the inability to concentrate and significantly lower cognitive function one year after acute dengue diagnosis [11]. Muscle pain, arthralgia and general malaise, retro-ocular pain, headache, nausea, and vomiting, are some of the other reported chronic symptoms [11]. Female sex, advanced age, presence of chills, and absence of rash are associated with post-dengue fatigue. Alterations in the immunological markers, such as C3 and C4 complement factors, C-reactive protein (CRP), and high IgG titers, are seen in some patients with post-dengue symptoms [12]. Furthermore, elevated levels of antinuclear antibodies (ANA) and immune complexes (IC) and FcγRIIa gene polymorphism have been implicated, suggesting an immune underpinning for the long-term persistence of dengue symptoms [13].

## 5. Predicting Severe Dengue

Improving risk prediction for severe dengue is crucial because of the uncertainty around which patients will enter the critical stage of dengue. The development of predictive markers for severe dengue will allow timely clinical management of shock and bleeding, permit early referrals, and facilitate the allocation of resources [14]. Reduced platelet count (until the platelet count is reduced to be considered dengue) and high aspartate transaminase in the first 72 h indicate a risk of severe dengue. Abdominal pain, vomiting, liver enlargement, and altered hyaluronan level also suggest progression to severe dengue [15]. The phenomenon of antibody-dependent enhancement (ADE) is known to mediate serious complications in secondary dengue infections. ADE ensues because of a mismatch between infecting serotype and the memory adaptive immunity developed from primary infection. ADE occurs due to the communication between sub-neutralizing DENV-reactive IgG antibodies and leucocytes, where the IgG Fc domain interacts with the Fcγ receptors on the leucocytes. DENV infection causes an increase in IgG1 afucosylation, and therefore, an elevated fraction of afucosylated anti-DENV IgG1 could be used as a prognostic tool to predict susceptibility to severe dengue [16]. In another recent research, high levels of 10 inflammatory and vascular biomarkers have been identified to be linked to severe outcomes in children and adults. The combination of IL-1RA, Ang-2, IL-8, ferritin, IP-10, and SDC-1 is associated with moderate to severe dengue in children, while the combination of SDC-1, IL-8, ferritin, sTREM-1, IL-1RA, IP-10, and sCD163 is associated with moderate to severe dengue in adults [17].

Viremia has been thought to be important in the dengue disease severity for a long time. Some previous studies indicated that higher viremia correlated to severer symptoms [18,19,20]. Meanwhile, no relationship between the viremia and disease severity was also reported by other groups [21,22,23]. The pathogenesis of severe dengue is attributed to the complex interplay between the virus, host genes, and host immune response. Therefore, viremia alone may not be sufficient, and other immunological factors may determine the progression to severe disease.

## 6. Dengue Vaccine Update

There is an unmet need for a multivalent dengue vaccine effective against all four DENV serotypes, which can be administered without pre-vaccination screening in both seropositive and seronegative individuals. V181 (live-attenuated, quadrivalent dengue virus vaccine) and TAK-003 (live-attenuated tetravalent dengue vaccine) are promising new vaccine candidates under development. The two V181 formulations (TV003 and TV005) increase DENV seropositivity to all four serotypes in both the baseline flavivirus-experienced (BFE) and baseline flavivirus-naïve (BFN) subgroups. At 6 months following the first dose, 92.6% of TV003 BFN, 74.2% of TV005 BFN, and 100% of TV003 and TV005 BFE participants demonstrated tri-or tetravalent DENV seropositivity [24]. TAK-003 has demonstrated overall vaccine efficacy of 80.2%, with vaccine efficacies of 76.1% and 66.2% in seropositive and seronegative individuals, respectively. TAK-003 vaccine efficacy against hospitalized dengue is reported to be 90.4%, and vaccine efficacy against dengue hemorrhagic fever is 85.9% in children aged 4–16 years [25]. In a phase two randomized controlled trial conducted in Panama, a dengue-endemic country, TAK-003 elicited cross-reactive (against DENV-1, -3, and -4 serotypes) and multi-functional (IFN-γ + TNF-α + IL-2+) CD4+ and CD8+ T-cell response regardless of pre-vaccination DENV neutralizing antibodies [26]. Furthermore, an ongoing long-term follow-up study shows that the efficacy of TAK-003 against symptomatic dengue is maintained over 3 years. The efficacy against virologically confirmed dengue (VCD) is 65.0% and 54.3%, and efficacy against hospitalized VCD is 86.0% and 77.1% in seropositive and seronegative individuals, respectively. Though the efficacy against VCD declined (44.7%) in the third year, the efficacy against hospitalized dengue was maintained (70.8%) [27]. Implementation strategies for the dengue vaccination programs have been developed, especially the school-based approach. However, various challenges include communication about limiting the use of this vaccine to seropositive individuals only. The significant barrier remains the cost of the vaccine and testing costs. The ability to be afforded will vary in each country, as will government policy and community acceptance [28]. To level up the accessibility of the vaccine by the population according to their socio-economic level, the dengue vaccine should be considered part of the national immunization program.

## 7. Advances in Dengue Therapeutic Targets

The dengue virus non-structural protein 1 [NS1] is a 48 kDa glycoprotein and is essential for viral replication. NS1 is continuously secreted by the infected host cells, at first as a monomer and later converted to homodimers following post-translational modification. NS1 is involved in the pathogenesis of vascular leakage through a complex interaction between the virus and the host immune response. NS1 promotes vascular leakage through disruption of the endothelial tight junctions, autophagy of endothelial cells, and degradation of the endothelial glycocalyx. NS1 also contributes to coagulopathy and thrombocytopenia, thus making it a potential dengue therapeutic and vaccine candidate [29]. NS1-induced platelet activation, aggregation, and over-destruction are implicated in the pathogenesis of coagulopathy during dengue infection. Due to its role in the development of vascular leak and coagulopathy, NS1 is a potential dengue therapeutic and vaccine candidate. NS1 not only anchors to the surface of the infected but is also secreted by the infected cells, thus allowing it to trigger both humoral and cellular immune responses, consequently making it a candidate for the dengue vaccine. The development of anti-NS1 antibodies, which can block all four dengue NS1 serotypes without cross-reacting with host proteins, could provide an attractive therapeutic target. 

Platelet-activating factor (PAF) is an inflammatory lipid mediator that is produced by mast cells, monocytes, and endothelial cells. High PAF levels are seen in patients with dengue hemorrhagic fever (DHF), increasing just before the onset of the critical phase, suggesting that PAF plays an important part in the pathogenesis of vascular leakage, thus making it a possible therapeutic target. PAF reduces endothelial integrity and mediates vascular leakage by reducing the expression of tight junction proteins (ZO-1) and trans-endothelial resistance (TEER). The potential role of PAF as a dengue therapeutic target has been demonstrated in human endothelial cell lines where PAF receptor blockers significantly inhibited the PAF downregulation of ZO-1 and TEER [30]. In a phase 2 trial in acute dengue patients in Sri Lanka, those receiving 40 mg of oral rupatadine (PAF receptor blocker) for 5 days were less likely to develop dengue hemorrhagic fever (DHF) (9.7%) compared to those receiving placebo (17.5%), although this difference was not statistically significant. Rupatadine also reduced the persisting vomiting, headache, and hepatic tenderness, and patients were less likely to develop a reduction in platelet counts <50,000 cells/mm^3^. It is postulated that rupatadine might be responsible for alleviating headache, vomiting, and hepatic tenderness by reducing the production of cytokines such as IL-6, TNFα, IL-1β, and mast cell products. Consequently, a combination of rupatadine with other drugs may have a significant benefit and should be investigated. These promising results encourage further research on the role of rupatadine in reducing the incidence of DHF [31]. Similarly, other inflammatory mediators, such as serum secretory phospholipase, urinary leukotrienes, vascular endothelial growth factor, and angiopoietin-2, among others, could be explored further as potential therapeutic targets.

## 8. Innovations in Dengue Control, Diagnostics, Therapeutics

The Wolbachia innovation is a promising method of controlling dengue transmission where the deliberate introduction of Wolbachia infection in Aedes aegypti male mosquitoes causes cytoplasmic incompatibility, thus interrupting dengue transmission [32]. The AWED trial (Applying Wolbachia to Eliminate Dengue) was a cluster-randomized controlled trial to assess the efficacy of Wolbachia-infected mosquito deployments in reducing the incidence of dengue in Yogyakarta City in Indonesia. The study involved two groups: 12 intervention clusters that received Wolbachia-infected A. aegypti deployments and 12 control clusters with no deployments. The primary endpoint of the AWED trial was symptomatic virologically confirmed dengue (VCD) due to any dengue serotype. The AWED trial (Applying Wolbachia to Eliminate Dengue) in Yogyakarta City, Indonesia, showed that Wolbachia deployments resulted in 77.1% protective efficacy against VCD (similar against four dengue virus serotypes) and 86.2% protective efficacy against hospitalized dengue [33]. This Wolbachia intervention has demonstrated a significant public health benefit and offers the added advantage of being self-sustaining as it does not require reapplication.

In another success story, a 3-year study conducted in Singapore showed that the Wolbachia-infected male-only release reduced wildtype Aedes aegypti population (by 92.7% in Yishun and 98.3% in Tampines, both residential estates in Singapore) and also reduced dengue incidence by 71–88% in high-rise urban target areas [34]. Singapore is the first country to implement the Wolbachia technique in a high-rise and high-density tropical environment. The Singapore National Environment Agency (NEA) plans to expand Project Wolbachia to cover 31% of all Housing and Development Board blocks and more than 300,000 households from July 2022 [35].

Talking about dengue diagnostics, both direct methods (detection of dengue virus and antigens, genomic sequencing) and indirect methods (detection of dengue-specific IgM, IgG, or IgA antibodies) require a blood sample. However, patient aversion to venipuncture, the need for trained personnel, lab equipment, and maintenance of cold chain for these diagnostic tests are significant barriers to dengue diagnosis and surveillance. The development of non-blood-based and non-invasive assays for quantifying DENV-specific antibodies might significantly simplify diagnostic tests and improve test acceptance. Non-invasive urine and saliva-based diagnostic methods are under development and could be used to strengthen vector control in areas with active dengue transmission [36].

Considering innovations in dengue vaccines, a new chimeric virus vaccine candidate has recently been developed using insect-specific flavivirus (ISF) named Binjari virus (BinJV). A preliminary study using a high-density microarray patch allowed the targeted delivery of a single dose of the vaccine and produced a potent immune response in the DENV mouse model [37]. Another advance in dengue therapeutics is the JNJ-1802, a pan-serotype anti-dengue small molecule currently undergoing further research. JNJ-1802 is a highly potent dengue virus inhibitor blocking the interaction between two viral proteins (NS3 and NS4B) and prevents the formation of the viral replication complex [38]. Other research advances, such as the development of dengue human infection models, could assist in predicting the efficacy of therapeutic and vaccine candidates, validating host biomarkers, and evaluating host responses in severe dengue [39]. 

## 9. Conclusions

The continuous evolution of the DENV serotype and genotype and their role in developing dengue outbreaks emphasize the need for consistent epidemiological surveillance for effective dengue control. The growing prevalence of NCDs and the epidemiological shift in dengue affecting older adults indicates that the dengue disease burden includes complex cases with underlying co-morbidities. Improving risk prediction for severe dengue is crucial as it will allow timely clinical management. The V181 and TAK-003 are promising new vaccine candidates under development. The development of anti-NS1 antibodies and PAF receptor blocker could provide attractive therapeutic targets. The Wolbachia intervention has demonstrated a significant public health benefit and offers the added advantage of being self-sustaining as it does not require reapplication. Non-invasive urine and saliva-based diagnostic methods are under development and could be used to strengthen vector control in areas with active dengue transmission. Considering innovations, a new chimeric virus vaccine candidate and the pan-serotype anti-dengue small molecule are currently undergoing further research. The 5th Asia Dengue Summit emphasized the immense public health burden of dengue in Asia and highlighted the need for strong multi-sectoral collaboration between health, research and innovation, environment, education, private sector, and communities to tackle the growing menace of dengue (Figure 1). 

## Figures and Tables

**Figure 1 tropicalmed-08-00231-f001:**
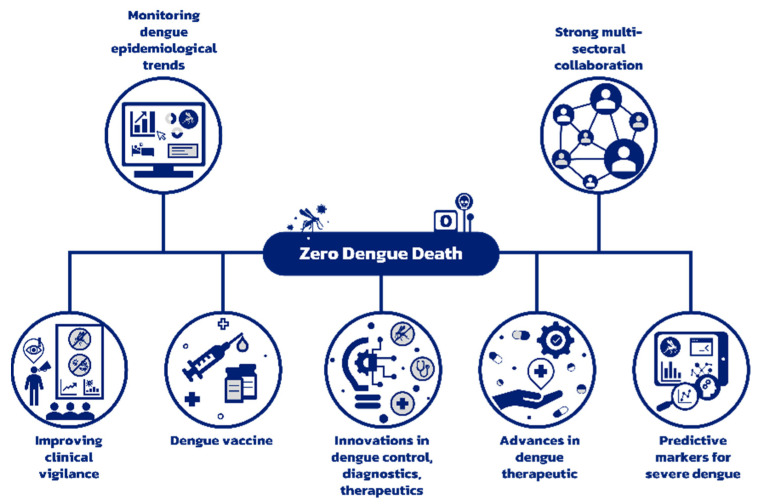
Strategy to reduce preventable dengue deaths to zero.

## Data Availability

Not applicable.

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
