# Peer review of "Proceedings of the 5th Asia Dengue Summit"

_tropicalmed, 2023, doi:10.3390/tropicalmed8040231_

Round 1

Reviewer 1 Report

The manuscript titled "Conference Report, Proceedings of the 5th Asia Dengue Summits" is clearly written and relevant to the field and presented in a well-structured manner. Although there is a lot of literature on dengue, the authors clearly highlight the current status of dengue, its epidemiology, vaccine development, detection methods, and innovations for its reduction. Accordingly, it is a valuable document that should be considered for publication.

Line 62. Can you express any percentage or rate?

Line 62. Add mortality data

Line 89. Counties or countries?

Line 135. Until the platelet count is reduced to be considered dengue

Line 161. 80.2?

Line 163. 90.4?

Line 164. 85.9?

Line 173. The authors mention the effectiveness of vaccines, but something that increases the importance of this document is that the authors add information about the accessibility of the vaccine by the population according to their socio -economic level.

Line 256. For the conclusions I suggest that you also add the most outstanding conclusion for each point: Monitoring dengue epidemiological trends, Improving clinical vigilance, Understanding chronic post-dengue sequelae, Predicting severe dengue, Dengue vaccine update, Advances in dengue therapeutic targets, Innovations in dengue control, diagnostics, therapeutics

Author Response

Reviewer 1

The manuscript titled "Conference Report, Proceedings of the 5th Asia Dengue Summits" is clearly written and relevant to the field and presented in a well-structured manner. Although there is a lot of literature on dengue, the authors clearly highlight the current status of dengue, its epidemiology, vaccine development, detection methods, and innovations for its reduction. Accordingly, it is a valuable document that should be considered for publication.

  1. Line 62. Can you express any percentage or rate? Line 62. Add mortality data

Response We are grateful for this comment. We have added the suggested information on Page 2, Line 61-62, as the following sentences.

Asia bears 70% of global dengue burden [1] with Southeast Asia recording the highest age-standardised incidence rates with an annual average of 34.3 cases per 1000 people and highest dengue mortality rates (8.49 cases per million in 2013) [2].

  1. Line 89. Counties or countries?

Response Thank you for pointing this out. We have revised the “counties” to “countries”  on Page 3, Line 86.

  1. Line 135. Until the platelet count is reduced to be considered dengue

Response Thank for comments. We have added the suggested information on Page 4, Line 132 as the following sentences.

Reduced platelet count (until the platelet count is reduced to be considered dengue) and high aspartate transaminase in the first 72 hours indicate risk of severe dengue.

  1. Line 161. 80.2? Line 163. 90.4? Line 164. 85.9?

Response We thank the reviewer for pointing out our mistake. It has been corrected on Page 5, Line 162, 164, and 165.

  1. Line 173. The authors mention the effectiveness of vaccines, but something that increases the importance of this document is that the authors add information about the accessibility of the vaccine by the population according to their socio -economic level.

Response Thank you for this suggestion. We have added the suggested information on Page 5, Line 173-180 as the following sentences.

Implementation strategies for the dengue vaccination programs have been developed, especially the school-based approach. However, various challenges include communication about limiting the use of this vaccine to seropositive individuals only. The significant barrier remains the cost of the vaccine and testing costs. The ability to be afforded will vary among each country, as will government policy and community acceptance [28]. To level up the accessibility of the vaccine by the population according to their socio-economic level, the dengue vaccine should be considered as part of the national immunization program.

  1. Line 256. For the conclusions I suggest that you also add the most outstanding conclusion for each point: Monitoring dengue epidemiological trends, Improving clinical vigilance, Understanding chronic post-dengue sequelae, Predicting severe dengue, Dengue vaccine update, Advances in dengue therapeutic targets, Innovations in dengue control, diagnostics, therapeutics

Response We have revised the conclusion on Page 7 as the following sentences.

Continuous evolution of DENV serotype and genotype and their role in developing dengue outbreaks emphasizes the need for consistent epidemiological surveillance for effective of dengue control. The growing prevalence of NCDs and the epidemiological shift in dengue affecting older adults indicates that the dengue disease burden includes complex cases with underlying co-morbidities. Improving risk prediction for severe dengue is crucial as it will allow timely clinical management. The V181 and TAK-003 are promising new vaccine candidates under development.  The development of anti-NS1 antibodies and PAF receptor blocker could provide attractive therapeutic targets. The Wolbachia intervention has demonstrated a significant public health benefit and offers an added advantage of being self-sustaining as it does not require reapplication. Non-invasive urine and saliva based diagnostic methods are under development and could be used to strengthen vector control in areas with active dengue transmission. Considering innovations, a new chimeric virus vaccine candidate and pan-serotype anti-dengue small molecule currently undergoing further research. The 5th Asia Dengue Summit emphasized the immense public health burden of dengue in Asia and highlighted the need for strong multi-sectoral collaboration between health, research and innovation, environment, education, private sector, and communities to tackle the growing menace of dengue (Figure 1).

Reviewer 2 Report

Overall, this manuscript is well written. I think that even people, who could not catch the summit, are able to understand what topics were discussed there. I suggest some points to be clarified.

Major comment

5. Predicting severe dengue

Viremia has been thought to be important in the dengue disease severity for a long time. Some previous studies indicated that the higher viremia correlated to the severer symptoms [PMID: 21909448, 10608744, 33340040]. Meanwhile, no relationship between the viremia and disease severity was also reported by other groups [PMID: 18269315, 26982706, 27986543]. Authors need to mention this point.

Minor comment

6. Dengue vaccine update

Please unify the order of description, “seropositive and seronegative” or “seronegative and seropositive”. Then, it will be helpful for readers to grasp easily the vaccine efficacy.

Author Response

Reviewer 2

Overall, this manuscript is well written. I think that even people, who could not catch the summit, are able to understand what topics were discussed there. I suggest some points to be clarified.

  1. Major comment 5. Predicting severe dengue. Viremia has been thought to be important in the dengue disease severity for a long time. Some previous studies indicated that the higher viremia correlated to the severer symptoms [PMID: 21909448, 10608744, 33340040]. Meanwhile, no relationship between the viremia and disease severity was also reported by other groups [PMID: 18269315, 26982706, 27986543]. Authors need to mention this point.

Response We think this is an excellent suggestion. We have added the suggested content to the manuscript on Page 4-5, Line 147-152 as the following sentences.

Viremia has been thought to be important in the dengue disease severity for a long time. Some previous studies indicated that the higher viremia correlated to the severer symptoms [18-20]. Meanwhile, no relationship between the viremia and disease severity was also reported by other groups [21-23]. The pathogenesis of severe dengue is attributed to the complex interplay between the virus, host genes, and host immune response. Therefore, viremia alone may not be sufficient, and other immunological factors may determine the progression to severe disease.

  1. Minor comment 6. Dengue vaccine update. Please unify the order of description, “seropositive and seronegative” or “seronegative and seropositive”. Then, it will be helpful for readers to grasp easily the vaccine efficacy.

Response As suggested, we have unified the order of description to “seropositive and seronegative” throughout the Dengue vaccine update section on Page 5.

Reviewer 3 Report

The paper describes conclusions drawn from The 5th Asia Dengue Summit held in Singapore from 13th-15th June 2022. The manuscript is well formulated and well prepared. It explains to the reader the state of knowledge concerning dengue epidemiological trends as well as innovations in dengue control, diagnostics and therapeutics. References are generally relevant and referenced correctly. In my opinion at least one illustration would improve attractiveness of the paper, but generally the paper can be accepted in present form.

Author Response

Reviewer 3

The paper describes conclusions drawn from The 5th Asia Dengue Summit held in Singapore from 13th-15th June 2022. The manuscript is well formulated and well prepared. It explains to the reader the state of knowledge concerning dengue epidemiological trends as well as innovations in dengue control, diagnostics and therapeutics. References are generally relevant and referenced correctly. In my opinion at least one illustration would improve attractiveness of the paper, but generally the paper can be accepted in present form.

Response We are grateful for this comment. We have added Figure 1. “Strategy to reduce preventable dengue deaths to zero” to the manuscript as suggested.

Figure 1. Strategy to reduce preventable dengue deaths to zero.
